# *Guy1*, a Y-linked embryonic signal, regulates dosage compensation in *Anopheles stephensi* by increasing X gene expression

Yumin Qi[1,2†], Yang Wu[3†], Randy Saunders[1,2], Xiao-Guang Chen[2], Chunhong Mao[4‡], James Kite Biedler[1,2]*, Zhijian Jake Tu[1,2]*

[1]Department of Biochemistry, Virginia Tech, Blacksburg, Virginia, United States; [2]Fralin Life Science Institute, Virginia Tech, Blacksburg, Virginia, United States; [3]Department of Pathogen Biology, School of Public Health, Southern Medical University, Guangdong, China; [4]Biocomplexity Institute of Virginia Tech, Virginia Tech, Blacksburg, Virginia, United States

*For correspondence:
jbiedler@vt.edu (JKB);
jaketu@vt.edu (ZJT)

†These authors contributed equally to this work

Present address: ‡Biocomplexity Institute & Initiative, University of Virginia, Charlottesville, VA, United States

Competing interests: The authors declare that no competing interests exist.

**Abstract** We previously showed that *Guy1*, a primary signal expressed from the Y chromosome, is a strong candidate for a male-determining factor that confers female-specific lethality in *Anopheles stephensi* (Criscione et al., 2016). Here, we present evidence that *Guy1* increases X gene expression in *Guy1*-transgenic females from two independent lines, providing a mechanism underlying the *Guy1*-conferred female lethality. The median level gene expression (MGE) of X-linked genes is significantly higher than autosomal genes in *Guy1*-transgenic females while there is no significant difference in MGE between X and autosomal genes in wild-type females. Furthermore, *Guy1* significantly upregulates at least 40% of the 996 genes across the X chromosome in transgenic females. *Guy1*-conferred female-specific lethality is remarkably stable and completely penetrant. These findings indicate that *Guy1* regulates dosage compensation in *An. stephensi* and components of dosage compensation may be explored to develop novel strategies to control mosquito-borne diseases.
DOI: https://doi.org/10.7554/eLife.43570.001

## Introduction

Sex-determination by a pair of heteromorphic X and Y sex chromosomes is one of a diverse array of mechanisms used by sexually reproducing organisms. The Y chromosome is often sparse in genes while the X retains hundreds or thousands of genes. Thus, monosomy of the relatively gene-rich X chromosome in the heterogametic sex (XY) poses a dosage problem that needs to be resolved. Diverse mechanisms have evolved to address the sex chromosome gene dosage imbalance (*Mank, 2013*; *Disteche, 2016*; *Graves, 2016*; *Gu and Walters, 2017*; *Lucchesi, 2018*; *Samata and Akhtar, 2018*). In *Drosophila melanogaster*, to compensate for the X monosomy in males, a mechanism known as complete dosage compensation works by hyper-expressing the entire X chromosome to equalize gene expression with that of females having two X chromosomes.

Dosage compensation in Drosophila requires three critical components: sex specificity, X chromosome specificity, and chromosome-wide up-regulation of transcription. The X chromosome-wide up-regulation of transcription is mediated by a dosage compensation complex (DCC), or more specifically the Male-Specific Lethal (MSL) complex. The MSL complex is a ribonucleoprotein comprised of 5 protein subunits (MSL-1, MSL-2, MSL-3, MOF, MLE) and a lncRNA (*Franke and Baker, 1999*; *Samata and Akhtar, 2018*). Of all the protein subunits, it is only MSL-2 that has sex-specific

expression and is restricted to males (*Bashaw and Baker, 1995*; *Kelley et al., 1995*). While transcribed in females, *msl-2* translation is prevented by multiple mechanisms, all involving the female-specific *sex-lethal* (*sxl*) protein which is also the primary signal of sex-determination in *Drosophila*. Thus, the female sex-determination signal SXL also acts to inhibit MSL-2 expression and therefore suppresses dosage compensation in females. X-chromosome targeting by the MSL complex involves binding to what are called high-affinity sites (HAS), particularly sites that contain a 21 base GA-rich motif (*Villa et al., 2016*). Initial binding to these sites is followed by spreading of the MSL complex to sites associated with actively transcribed genes. The MSL complex is essentially a chromatin-modifying complex using the MOF protein subunit, a histone acetyltransferase, to acetylate histone 4 lysine 16 (H4K16ac), resulting in a chromatin conformation conducive to upregulation of X-linked genes (*Smith et al., 2000*; *Smith et al., 2001*).

As mentioned above, the master switch of sex-determination *sxl* also regulates dosage compensation in *Drosophila* (*Schütt and Nöthiger, 2000*). Loss-of-function *sxl* mutants cause female embryonic lethality in *D. melanogaster*, most likely due to mis-regulation of dosage compensation, thereby upregulating or overexpressing X chromosome genes in females (*Cline, 1978*). In other model species *Bombyx mori* and *Caenorhabditis elegans*, the master switches of sex-determination *Fem/Masc* and *xo-lethal 1*, also regulate dosage compensation, respectively, and loss of function of these genes results in sex-specific lethality (*Miller et al., 1988*; *Kiuchi et al., 2014*).

In Anopheline mosquitoes which have heteromorphic sex chromosomes, dosage compensation might be expected in the heterogametic sex, and dosage compensation was indicated for *An. gambiae* based on the analysis of published microarray data (*Baker et al., 2011*; *Baker and Russell, 2011*; *Mank, 2013*). Indeed, complete dosage compensation was recently demonstrated in *An. stephensi* and *An. gambiae* based on RNA-Seq analysis (*Jiang et al., 2015*; *Rose et al., 2016*). Complete dosage compensation has also recently been demonstrated for members of the *An. gambiae* species complex (*Deitz et al., 2018*). As described above, sex-determination and dosage compensation have been shown to be tightly linked. Y chromosome genes and M-factor candidates *Guy1* and *gYG2/Yob*, confer female lethality in *An. stephensi* and *An. gambiae*, respectively (*Criscione et al., 2016*; *Krzywinska et al., 2016*) consistent with the hypothesis that mis-regulation of dosage compensation is responsible. In this work, we provide direct evidence that ectopically expressed *Guy1* upregulates X-linked gene transcription in females, resulting in an imbalance of gene dosage and is the most likely cause of female-specific lethality. These findings directly link an embryonic signal expressed from a mosquito Y chromosome to the regulation of dosage compensation and support a mechanism of hyper-expression of X chromosome genes. Moreover, the stability and penetrance of the female-specific lethality conferred by the *Guy1* transgene suggests that dosage compensation may be explored to develop novel strategies to control mosquito-borne diseases.

## Results

### Transgenic lines expressing *Guy1* have produced male-only offspring for up to 60 generations

The transgenic *Guy1*-expressing lines used in this study have produced only transgenic male offspring for a cumulative 77 generations, consistent with previous results using other transgenic *Guy1*-expressing lines (*Criscione et al., 2016*). The nGuy1_2 line which expresses *Guy1* utilizing its native promoter and 3' UTR (*Criscione et al., 2016*) has been reared for 60 generations as of July 2018, while the new bGuy1tC line which utilizes the bZip1 early zygotic promoter has been reared for 17 generations. The bGuy1tC line construct has a Twin C-terminal tag (*Schmidt et al., 2013*) to facilitate transgenic protein detection and isolation, which does not appear to affect the phenotype. Not a single transgenic female has been detected in 9615 screened progeny (*Table 1*). These results from two independent transgenic lines demonstrate the complete penetrance and stability of the *Guy1*-conferred female lethality.

**Table 1.** Adult screening of *Guy1*-expressing lines *nGuy1_2* and *bGuy1tC* demonstrate 100% female lethality.

| Line | TM (%) | TF | WM (%) | WF (%) | Total progeny |
|------|--------|----|--------|--------|---------------|
| *nGuy1_2* | 1970 (35.5) | 0 | 1800 (32.5) | 1776 (32.0) | 5546 |
| *bGuy1tC* | 1506 (37.0) | 0 | 1340 (32.9) | 1223 (30.1) | 4069 |
| | 3476 | 0 | 3140 | 2999 | 9615 |

TM stands for transgenic male and WM stands for wild type male. TF stands for transgenic female and WF stands for wild type female.

For these screening results line *nGuy1_2* was at generation 59, 60 and line *bGuy1tC* was at generation 16, 17.

DOI: https://doi.org/10.7554/eLife.43570.002

## Genotyping of 1st instar larvae from transgenic *Guy1*-expresssing lines indicates late transgenic female hatching and lethality

A small percentage of transgenic females are observed as L1 larvae but die shortly after hatching (*Criscione et al., 2016*). In order to obtain a better understanding of the nature and timing of the female lethality, and to inform selection of time points for RNA-Seq, we observed L1 larvae from several time points (*Table 2*) that were not sampled previously (*Criscione et al., 2016*). We took advantage of our ability to identify all four possible genotypes, transgenic and non-transgenic (hereafter called wild type) male and female siblings, using the same strategy as done previously using two transgenic markers, one in the transgenic cassette and one in the X chromosome (*Criscione et al., 2016*). The majority of wild-type males and females, and transgenic males, hatched by 42 hr post-oviposition as hatching normally occurs. However, no transgenic females were observed until 58 hr post-oviposition, and they died within 24 hr. Furthermore, the number of transgenic females were only about 10% of the total expected transgenic females. These results indicate that the majority of transgenic females die at the embryonic stage or during/shortly after hatching, and some transgenic females hatch later than normally expected but die shortly afterwards.

## The *Guy1* transgene acts in dosage compensation and broadly upregulates X-linked genes in transgenic females

We proposed that mis-regulation of dosage compensation may be responsible for the observed transgenic female lethality (*Criscione et al., 2016*) because this has been implicated in other insects (mentioned above). The few late-hatching transgenic females surviving as L1 larvae offered an opportunity to examine gene expression on a genome-wide scale using RNA-Seq. We performed two independent experiments using L1 larvae: one experiment included four biological replicates

**Table 2.** Monitoring 1st instar larvae indicates delayed transgenic female hatching and female lethality.

| Transgenic line | Genotype | 42 hr PO | 50 hr PO | 58 hr PO | 66 hr PO | 70 hr PO | 82 hr PO | Adults (%) |
|---|---|---|---|---|---|---|---|---|
| *nGuy1_2* | TF | 0 | 0 | 22 | 20 | 2 | 0 | 0 |
| | WF | 169 | 199 | 201 | N/A | N/A | N/A | 168 (32.0) |
| | TM | 181 | 206 | 209 | N/A | N/A | N/A | 175 (33.3) |
| | WM | 189 | 219 | 222 | N/A | N/A | N/A | 182 (34.7) |
| *bGuy1tC* | TF | 0 | 0 | 16 | 16 | 12 | 0 | 0 |
| | WF | 159 | 166 | 166 | N/A | N/A | N/A | 122 (33.2) |
| | TM | 162 | 169 | 170 | N/A | N/A | N/A | 136 (37.0) |
| | WM | 139 | 154 | 154 | N/A | N/A | N/A | 110 (29.9) |

TF stands for transgenic female and WF stands for wild type female. TM stands for transgenic male and WM stands for wild-type male. PO stands for Post Oviposition.

Numbers from 42 h – 82 h PO represent larval counts.

N/A - after 58 hr all larvae had hatched and only transgenic female larvae were monitored for death.

Similar experiments have been repeated more than three times for each line with essentially the same results.

DOI: https://doi.org/10.7554/eLife.43570.003

each for female transgenic and wild-type samples from the *nGuy1_2* line, and the other experiment included three biological replicates each for all four genotypes (male and female transgenic and wild type) from the *nGuy1_1* line (*Criscione et al., 2016*). Spearman correlation analyses of gene expression of biological replicates are shown in *Supplementary file 1* and PCA analyses are shown in *Figure 1—figure supplement 1*.

L1 larvae gene expression for *nGuy1_2* transgenic and wild-type females is depicted in Violin Plots as log2(FPKM +1) (*Figure 1*, data shown in *Supplementary file 2*). In transgenic females, the median gene expression of X-linked genes is significantly higher than that of the autosomes (53.6 vs. 36.6, p=$2.2e^{-16}$), whereas there is no significant difference found in wild-type females (35.4 vs. 36.9, p=0.31). One of the four transgenic female replicates (TF1) was excluded from the above analysis on the basis of PCA (*Figure 1—figure supplement 1A*). TF1 had a lower overall gene expression than the other replicates (*Supplementary file 3*). However, as shown in *Table 3* and *Supplementary file 3*, the median gene expression of X-linked genes is significantly higher than that of the autosomes in all four replicates including TF1 (38.1 vs. 33.4, p=0.0012), TF2 (54.3 vs. 36.2, p<$e^{-7}$), TF3 (53.0 vs. 36.1, p<$e^{-7}$), and TF4 (54.1 vs. 34.7, p<$e^{-7}$).

An independent experiment using three biological replicates each of all four genotypes including males from line *nGuy1_1* yielded similar results for transgenic females either when analyzed as a group (*Figure 1—figure supplement 2*) (median expression of 44.7 vs. 31.6, p=$5.4e^{-12}$, *Supplementary file 2*) or as individual replicates (*Supplementary file 3*). In this experiment, in males, no significant difference in median gene expression between X-linked and autosomal genes was observed (*Figure 1—figure supplement 2*, *Supplementary file 2*).

In order to quantify the number of differentially expressed genes on the X chromosome vs. the autosomes, and reveal any bias in X-linked gene upregulation, we used the DESeq2 R package (*Love et al., 2014*) to compare gene expression between *Guy1* transgenic and wild-type groups (see *Supplementary file 4* for DESeq2 Output). In transgenic females, there were 580 and 475 up-regulated genes on the X chromosome in two independent experiments, compared to 27 and 12 down-regulated genes, respectively (*Figure 2*, *Figure 2—figure supplement 1A*, *Supplementary files 4* and *5*). When considering the number of upregulated to down-regulated genes on the X chromosome vs. the autosomes, there is a clear bias toward upregulated X-linked genes in transgenic females for the two independent experiments (p=$2.2e^{-16}$, p=$2.2e^{-16}$) (*Supplementary file 5*). The ratio of upregulated to downregulated genes is ~68 and~29 times greater for the X chromosome compared to the autosomes in these two experiments, respectively. This bias is not seen in males (*Supplementary file 5*). In fact, this value is 0.69, suggesting that Guy1 may preferentially albeit slightly downregulate X-linked genes in transgenic males compared to wild-type males (p=0.0217). Of the 580 and 475 upregulated X-linked genes identified in the two experiments (*Supplementary file 5*), 382 genes were found in common between the two data sets (*Supplementary file 4*) and they encompass a broad functional range according to GO analysis

**Table 3.** The median level gene expression of genes on X chromosome and autosomes of individual replicates of line *nGuy1_2*.

| Sample* | Chromosome X | Autosomes | P value† |
|---|---|---|---|
| TF1 | 38.12126 | 33.37074 | 0.0012021 |
| TF2 | 54.33247 | 36.23512 | 0.0000000 |
| TF3 | 53.00118 | 36.08971 | 0.0000000 |
| TF4 | 54.14014 | 34.74229 | 0.0000000 |
| WF1 | 38.07142 | 38.49541 | 0.7098231 |
| WF2 | 36.15963 | 36.88966 | 0.6335401 |
| WF3 | 33.74510 | 35.75736 | 0.2684612 |
| WF4 | 30.98920 | 32.89602 | 0.0860183 |

*TF stands for transgenic female and WF stands for wild type female. Transcripts under a relatively low expression level (FPKM <1) are excluded. Results of different FPKM cutoffs are provided in Supplemental File S3.

†The p values were calculated based on the two-tailed two-sample Wilcoxon rank sum test.

DOI: https://doi.org/10.7554/eLife.43570.007

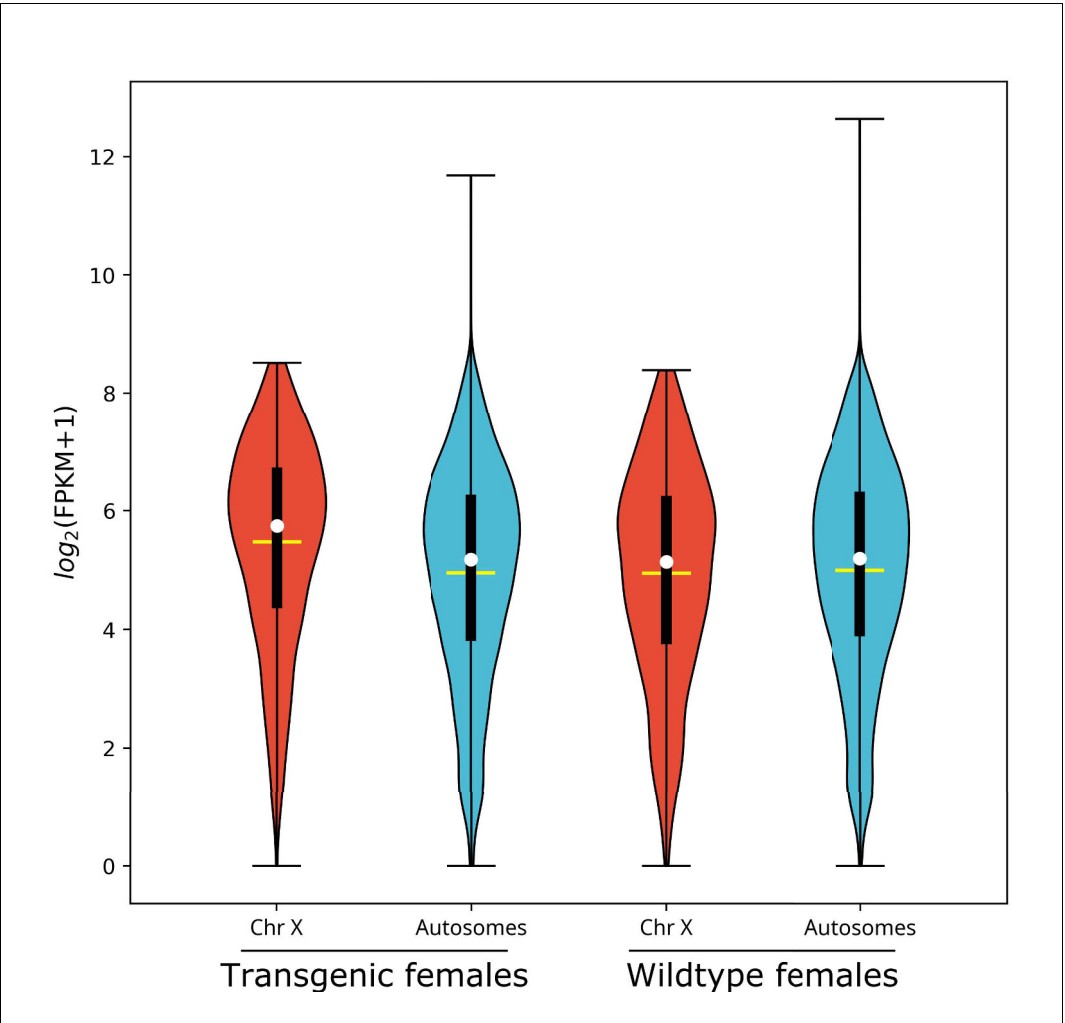

**Figure 1.** X chromosome median gene expression is up-regulated in *Guy1* transgenic females of line *nGuy1_2*. Violin plot shows gene expression as log2(FPKM +1) from the X chromosome and autosomes from transgenic and wild-type females. Transcripts under a relatively low expression level (FPKM <1) are excluded. The width of the violin plots shows the density of transcripts at different expression values. The white dot denotes the median of each group, while the yellow horizontal bar denotes the mean. The bottom and top of the vertical thick black bars represent the first and third quartiles, respectively. In transgenic females the median expression level of the X chromosome is significantly higher than the autosomes according to the two-tailed Wilcoxon tests, while no significant difference is seen in wild-type females. Median expression value comparisons between the X chromosome and autosomes for transgenic and wild-type females are 53.6 vs. 36.6 (p=2.2e$^{-16}$), 35.4 vs. 36.9 (p=0.31), respectively. One of the four transgenic female replicates (TF1) was excluded from the above analysis on the basis of PCA (*Figure 1—figure supplement 1*). TF1 had a lower overall gene expression than the other replicates (*Supplementary file 3*). However, the median gene expression of X-linked genes is still significantly higher than that of the autosomes in TF1 (*Table 3*). An independent experiment validated these results in females, and for additional male samples included, there was no significant difference in the median gene expression between the X chromosome and autosomes in transgenic males or non-transgenic males (*Figure 1—figure supplement 2*). All median FPKM expression value comparisons with statistical significance can be found in *Supplementary file 2*.

DOI: https://doi.org/10.7554/eLife.43570.004

The following figure supplements are available for figure 1:

**Figure supplement 1.** Principal component analysis of replicates from Experiment A (Panel A, two genotypes from *nGuy1_2*) and Experiment B (Panel B, four genotypes from *nGuy1_1*).

DOI: https://doi.org/10.7554/eLife.43570.005

**Figure supplement 2.** X chromosome median gene expression is up-regulated in *Guy1* transgenic females from line *nGuy1_1*.

*Figure 1 continued*

DOI: https://doi.org/10.7554/eLife.43570.006

(*Figure 3*). These 382 genes represent approximately 40% of the genes on the X chromosome and they are broadly distributed across the X chromosome (*Figure 4*). These results taken together, present strong evidence that *Guy1* plays a role in dosage compensation by upregulating the expression of X-linked genes.

# Discussion

Here, we provide evidence for potential link between sex-determination and dosage compensation. Extending from our previous work (*Criscione et al., 2016*), we generated multiple independent transgenic lines to ectopically express the *An. stephensi* M-factor candidate *Guy1* from an autosome and were able to demonstrate a female-lethal phenotype with remarkable stability and complete penetrance.

Furthermore, we used RNA-Seq to establish that *Guy1* is responsible for the upregulation of X-linked genes in transgenic females, and therefore is an initial regulator of dosage compensation. In other words, we have shown for the first time in a mosquito species that a primary embryonic signal expressed from the Y chromosome regulates dosage compensation by up-regulation of X-linked genes. However, it remains to be seen whether *Guy1* suppression in males will result in a decreased expression of X-linked genes. Challenges remain to achieving and verifying *Guy1* suppression as *Guy1* transcription is narrowly restricted to the early embryonic stage. The 382 (~40%) upregulated X-linked genes encompass a broad functional range (*Figure 3*) and are widely distributed across the

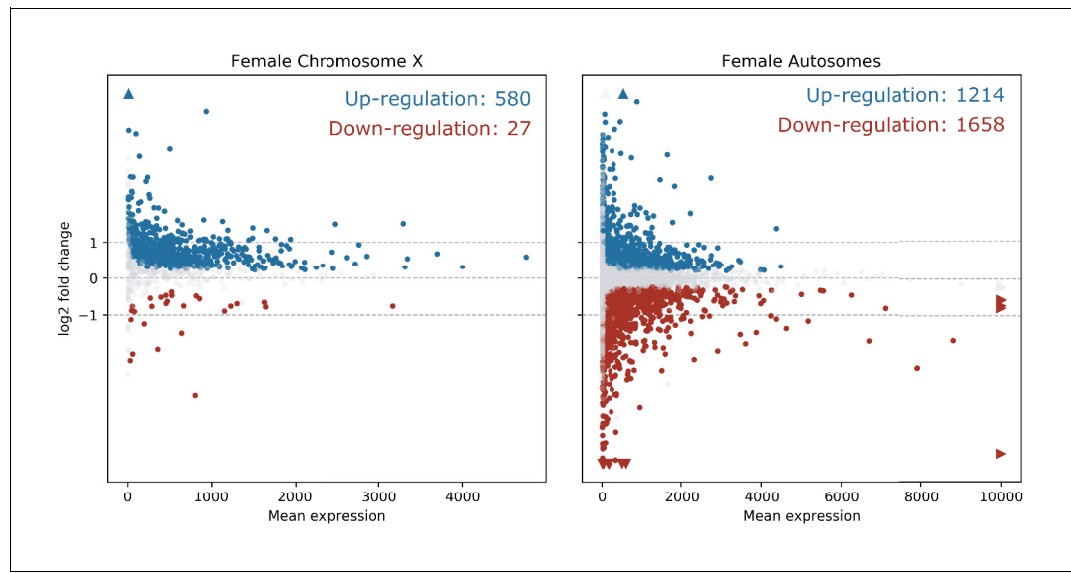

**Figure 2.** X chromosome genes are upregulated in *Guy1* transgenic females of line *nGuy1_2*. MA Plots show differentially expressed genes on the X chromosome and the autosomes in *Guy1* transgenic female compared to wild-type female siblings. Each dot represents one gene. Blue dots denote upregulated genes, the red dots denote downregulated genes, and grey dots are not differentially expressed (BH-adjusted p>0.1). Colored arrows indicate values beyond the range of the plot. Genes are significantly upregulated on the X chromosome in transgenic females compared to the autosomes, which is supported by a Chi-square test (p=2.20e$^{-16}$, *Supplementary file 5*). A repeat experiment with essentially the same results that also includes male samples is shown in *Figure 2—figure supplement 1*.

DOI: https://doi.org/10.7554/eLife.43570.008

The following figure supplement is available for figure 2:

**Figure supplement 1.** X-linked genes are upregulated in *Guy1* transgenic females from line *nGuy1_1*.

DOI: https://doi.org/10.7554/eLife.43570.009

X chromosome (*Figure 4*). In transgenic males which have an extra copy of *Guy1*, there was not an observable phenotype or significant alteration in the X/A expression ratio, suggesting the dosage compensation mechanism is not dose-responsive with regard to *Guy1* copy number. Dosage compensation is well-established in diverse organisms including Drosophila, nematodes, and vertebrates, and it has been shown in many cases that dosage compensation involves chromatin modification as a common theme to equalize expression in the heterogametic sex (*Kiuchi et al., 2014*; *Strome et al., 2014*; *Marin et al., 2017*; *Richard et al., 2017*; *Davis et al., 2018*). Therefore, it reasons that *Guy1* plays a role either directly or indirectly in affecting this epigenetic system on a chromosome-wide basis as for Drosophila. Mosquito orthologs to the Drosophila MSL complex proteins have been reported (*Zdobnov et al., 2002*; *Behura et al., 2011*; *Rose et al., 2016*) but homologs to *msl-1* were reported missing from *Ae. aegypti* and *An. gambiae* but present in *Cu. quinquefasciatus*, while homologs to *msl-2* were found in all three of these species (*Behura et al., 2011*). BLAST using *D. melanogaster* protein sequences performed against *An. stephensi* databases (Vectorbase. org) shows the MOF and MLE homologs having the highest conservation, the MSL-3 homolog being highly divergent, and no significant matches for MSL-1 or MSL-2. Whether any of these homologs

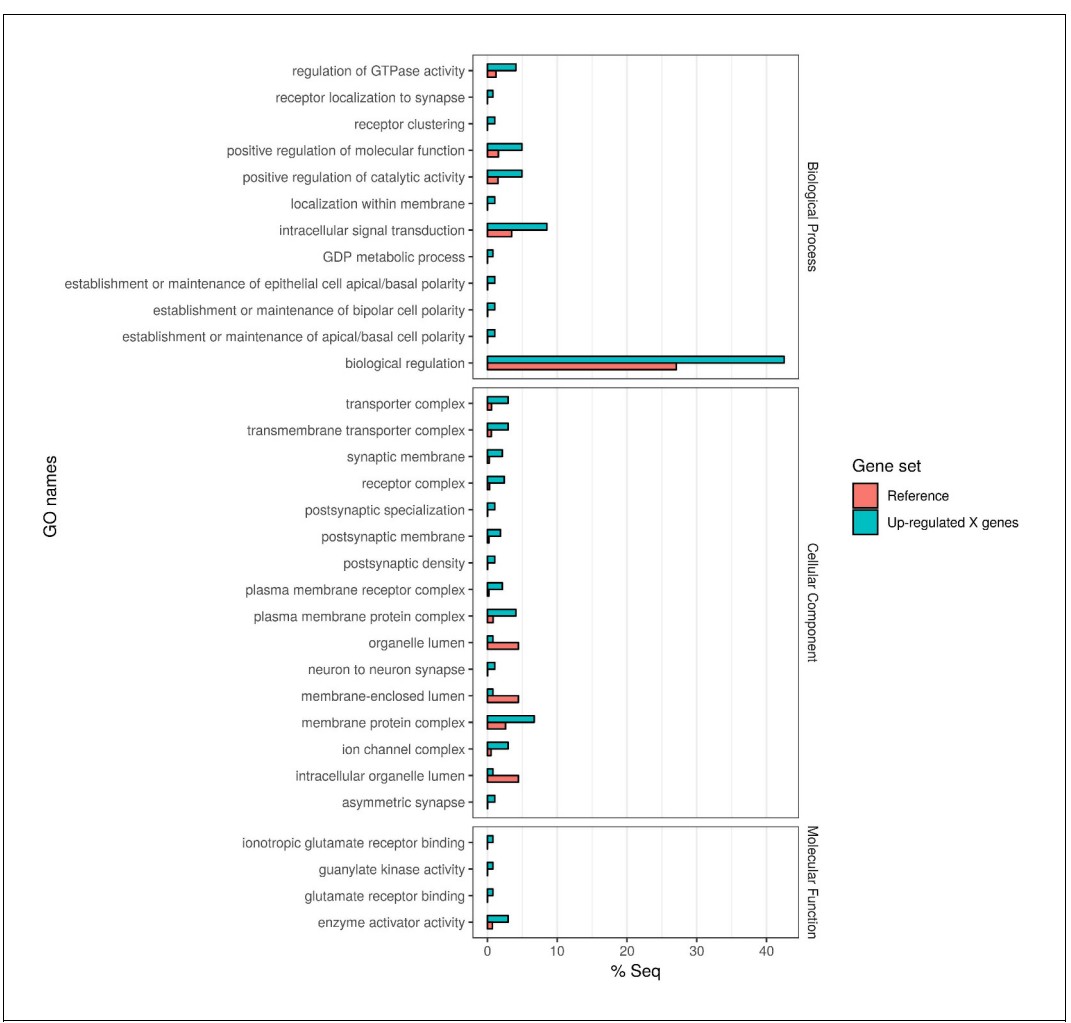

**Figure 3.** Gene Ontology enrichment of the 382 X chromosome genes that are upregulated in the *Guy1*-transgenic females from both *nGuy1_1* and *nGuy1_2* lines. There are 475 and 580 upregulated X-linked genes identified by DESeq2 in the transgenic females from line *nGuy1_1* and *nGuy1_2*, respectively (*Supplementary file 4* and *Supplementary file 5*). Among these, 382 genes were found in common (*Supplementary file 4*). These 382 genes were mapped to GO names and a two-tailed Fisher's exact test was performed to detect enrichment of GO names against AsteI2.2 transcripts under FDR of 0.05 using Blast2GO (V5.1.1) (*Götz et al., 2008*).
DOI: https://doi.org/10.7554/eLife.43570.010

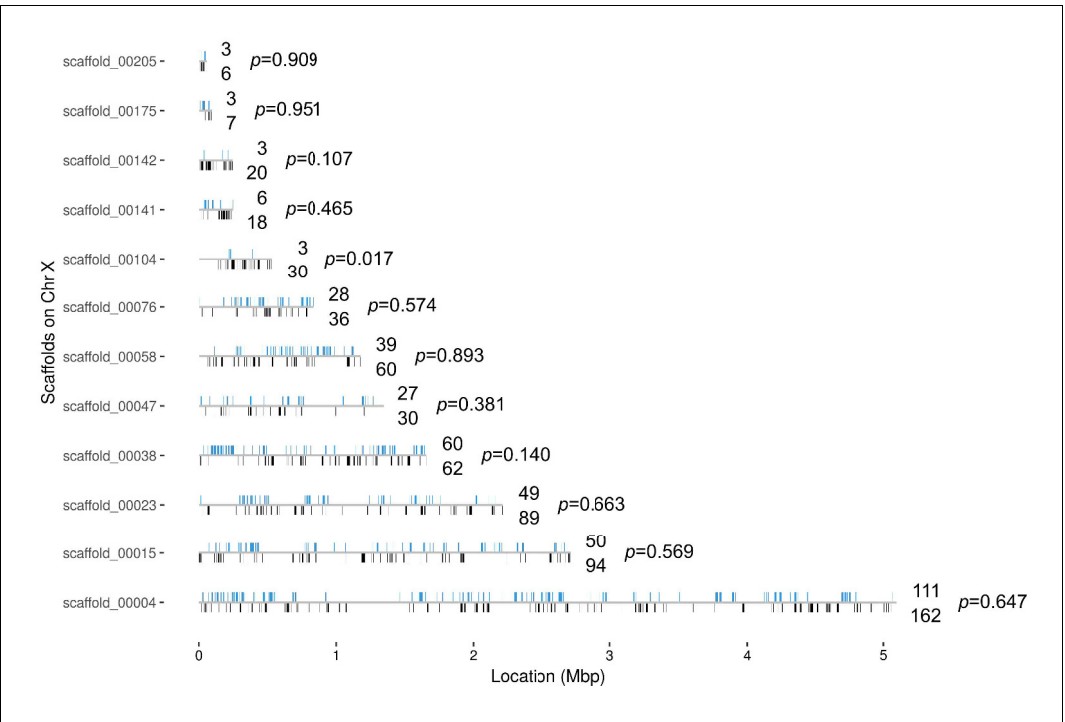

**Figure 4.** Distribution of the 382 X chromosome genes that are upregulated in the *Guy1*-transgenic females from both *nGuy1_1* and *nGuy1_2* lines. There is not yet a chromosomal level assembly for *Anopheles stephensi*. The location (from start to end) of the 996 X genes are mapped on the 12 X-scaffolds (***Supplementary file 4***). Blue bars above the line indicates the position of genes that are upregulated and the number above the line indicates the number of upregulated genes in each scaffold. Black bars underneath the line indicates the position of other genes and the number below the line indicates the number of other genes in each scaffold. There are only seven X genes that are down-regulated in the *Guy1*-transgenic females from both lines (***Supplementary file 4*** sheet 4). The p-values show the results of a Chi-square test of each scaffold for deviation from the 382/996 ratio. We also broke scaffolds 00023, 00015, and 00004 into 2, 3, and 5 evenly divided segments, respectively, to perform the same Chi-square test. Segment 2 of scaffold_00004 showed a p-value of 0.09 while all other nine segments showed p-values between 0.34 and 0.99. Thus, although minor local biases exist, the 382 upregulated genes are widely distributed across the X chromosome.
DOI: https://doi.org/10.7554/eLife.43570.011

play a role in mosquito dosage compensation remains to be determined. A chromatin-remodeling mechanism or dosage compensation complex must use similar machinery to alter chromatin architecture and promote transcriptional up-regulation on a chromosome-wide scale. Going forward, major questions to be addressed are: what is the mechanism of X chromosome gene upregulation in males and how is *Guy1* involved in this mechanism? Sequence analysis of the predicted 56 amino acid Guy1 protein suggests that it may be a DNA-binding protein and it could potentially participate in protein-protein interactions (***Criscione et al., 2013***; ***Criscione et al., 2016***). It is possible that Guy1 or a Guy1-containing protein complex could bind the X chromosome directly to exert its effect. However, it is perhaps more likely that Guy1 serves as the initial signal for dosage compensation and downstream effector(s) is required to confer or sustain X upregulation, as *Guy1* transcription is restricted to the early embryonic stage.

We have provided evidence for a mechanism that underlies the remarkably stable and penetrant female-specific lethality of the *Guy1* transgene, namely the up-regulation of the X-linked genes in the females which already have two X chromosomes. Improved understanding of the fundamental biological mechanisms of sex-determination and dosage compensation will inform the development of novel genetic-based strategies for the control of mosquito-borne diseases, which may include sex-separation for male selection in sterile insect technique, or population suppression via induced female lethality. The *Guy1* transgene, with its female-lethal phenotype, stability and penetrance,

may be suitable for such strategies. Its effect may be enhanced using a CRISPR-Cas9 gene drive as previously proposed (*Biedler et al., 2015*) and recently demonstrated in *An. gambiae* caged populations (*Kyrou et al., 2018*). Thus, components of dosage compensation may be explored to develop novel strategies to control mosquito-borne diseases.

# Materials and methods

**Key resources table**

| Reagent type (species) or resource | Designation | Source or reference | Identifiers | Additional information |
|---|---|---|---|---|
| Gene (*Anopheles stephensi*) | *Guy1* | PMID: 23683123; PMID: 27644420 | JX174417; AFS60403 | |
| Genetic reagent (*Anopheles stephensi*) | X-linked CFP (cyan fluorescent protein) line | PMID: 20113372 | | |
| Biological sample (*Anopheles stephensi*, Indian) | The transcriptome of Guy1-transgenic Anopheles stephensi L1 instar (TaxID: 30069) | this paper | PRJNA503140 (NCBI Sequence Read Archive) | |
| Recombinant DNA reagent | bGuy1tC; nGuy1 | this paper; PMID: 27644420 | | |
| Recombinant DNA reagent | piggyBac transformation vector and helper | PMID: 11151299 | | |
| Sequence-based reagent | An. stephensi reference genome (Version AsteI2); An. stephensi transcripts (Version AsteI2.2) | PMID: 25244985; can also be accessed at Vectorbase.org | | |
| Sequence-based reagent | Strep-tag II | PMID: 17571060 | | |
| Commercial assay or kit | Quick-RNA Miniprep kit | Zymo Research | R1054 | |
| Software, algorithm | HISAT (V2.1.0) | PMID: 25751142 | | |
| Software, algorithm | SAMtools | PMID: 19505943 | | |
| Software, algorithm | MarkDuplicates | https://broadinstitute.github.io/picard/ | | |
| Software, algorithm | StringTie (V1.3.3) | PMID: 25690850 | | |
| Software, algorithm | Python script | this paper; https://gist.github.com/yangwu91/ | 79b74035465978e78d41af4236597ff1 | |
| Software, algorithm | Matplotlib Python Package | DOI: 10.1109/Mcse.2007.55 | | |
| Software, algorithm | GenomicAlignments R package | PMID: 23950696 | | |
| Software, algorithm | DESeq2 | PMID: 25516281 | | |

## Transgenic constructs and transgenic lines

The transgenic construct used to express *Guy1* using its native promoter and 3' UTR has been described as *nGuy1* (*Criscione et al., 2016*). The *nGuy1_1* and *nGuy1_2* lines in this study have the same construct but with a different insertion site. Another construct we used in this study to generate line *bGuy1tC* is similar to the construct that utilized the *bZip1* early zygotic promoter to generate line *bGuy1C* (*Criscione et al., 2016*) but has Twin C-terminal tags (*Schmidt et al., 2013*). Transgenic

lines were generated using the *piggyBac* transformation method as described previously (*Criscione et al., 2016*).

## Monitoring the time of Hatch and death of the *Guy1* transgenic females

In order to be able to identify and monitor all four genotypes (*Guy1*-expressing transgenic and sibling wild-type males and females), crosses were set up as done previously to obtain progeny having fluorescent markers corresponding to each genotype: DsRed/CFP (*Guy1* transgenic female), DsRed/- (*Guy1* transgenic male), -/CFP (wild type female), -/- (wild type male) (*Criscione et al., 2016*). Here, transgenic is used to refer to the *Guy1* transgene, not the CFP marker. Egg cups were placed into cages for 2 hr, then removed and allowed to incubate at 28°C. Time zero is when the egg cup was initially placed in the cage (oviposition) and times hereafter refer to time zero. Hatching was monitored at several time points starting shortly after when *An. stephensi* larvae normally hatch (approximately 42 hr post-oviposition). For the first time point at 42 hr, eggs and any hatched larvae were rinsed from the egg papers into a 15 ml conical tube and placed on ice for 10 min. First instar larvae (L1) were removed from the bottom using a pipette, placed on a wet filter paper in a petri dish and genotyped according to fluorescence, while the 15 ml conical tube was returned to incubation at 28°C. For subsequent time points at 50 hr and 58 hr, any hatched larvae were removed and genotyped. By 58 hr, all eggs had hatched. Subsequently, female larval death was monitored at 66 hr, 70 hr and 82 hr. As adults, male or female sex was confirmed. This experiment was separate than that used to collect L1 larvae for RNA-Seq (see below).

## Screening of *Guy1* transgenic lines

Eggs were hatched in water containing Sera Micron Fry Food with brewer's yeast. Larvae were reared to the pupal stage in water containing Purina Game Fish Chow. The pupae were picked up and screened for the presence of fluorescent marker and their sex were determined by the tail of pupae and the presence of testes. Pupae with different genotypes were placed into different cups and allowed to emerge. Pupae and adults were monitored every day until all larvae pupated and all adults emerged.

## RNA-Seq of *Guy1* L1 larvae

To ensure L1 instar of the appropriate age were collected, we monitored the hatching time of the four sibling genotypes (*Table 2*) and noticed a delay in hatching of the transgenic females, similar to what we observed previously for *Guy1* transgenic lines (*Criscione et al., 2016*). In addition, only a small fraction of *Guy1* transgenic females are observed to hatch, and they die within 24 hr of hatching. Thus, we collected all four genotypes within 4 hr after they hatched. We performed two RNA-Seq experiments; one included three biological replicates each of all four genotypes (line *nGuy1_1*), and the other included four biological replicates each of transgenic and wild-type females (line *nGuy1_2*). Each biological replicate contained approximately 10 pooled individuals. RNA was extracted using the Quick-RNA Miniprep kit (Zymo Research, Irvine, CA) according to the manufacturer's protocol. Library prep and Illumina sequencing of the four genotypes was performed at the Virginia Biocomplexity Institute on the Virginia Tech campus in Blacksburg, VA. Libraries were sequenced for 75 cycles by an Illumina NextSeq to produce 400 million single reads.

## RNA-Seq data analysis and statistical methods

RNA-Seq reads from four genotype groups, including female wild type, female transgenic, male wild type and male transgenic mosquito samples were aligned using HISAT (V2.1.0) (*Kim et al., 2015*) to the *An. stephensi* reference genome (Version AsteI2) (*Jiang et al., 2014*) separately. RNA-Seq datasets are deposited in NCBI with BioProject accession number PRJNA503140. The resulting BAM files were sorted and indexed by SAMtools (*Li et al., 2009*). MarkDuplicates from Picard tool kit (https://broadinstitute.github.io/picard/) was used to identify and remove PCR duplicates. StringTie (V1.3.3) (*Pertea et al., 2015*) was used to estimate the relative abundances of the reference transcripts (Version AsteI2.2), which were downloaded from VectorBase (*Giraldo-Calderón et al., 2015*). Transcript expression levels were estimated as Fragments Per Kilobase per Million mapped reads (FPKM). The FPKM-normalized expression matrix of the transcripts was generated from StringTie results by an in-house Python script (https://gist.github.com/yangwu91/

79b74035465978e78d41af4236597ff1) (*Wu, 2018*; copy archived at https://github.com/elifescien-ces-publications/stringtie_FPKM.py). Spearman correlation analyses (https://www.rdocumentation.org/packages/psych/versions/1.80.12/topics/corr.test) of gene expression of biological replicates were performed (*Supplementary files 1*) to assess correlation within and between genotypes and to test whether the premise for DESeq2/EdgeR analyses was met. PCA analyses (https://www. rdocumentation.org/packages/DESeq2/versions/1.12.3/topics/plotPCA) were also performed to assess independent clustering of different genotypes (*Figure 1—figure supplement 1*). Transcripts with FPKM values less than 1, that is, under a relatively low expression level, were removed for generating Violin Plots using Matplotlib Python Package (*Hunter, 2007*). The Two-sample Two-tailed Wilcoxon rank sum test was applied to identify whether the median gene expression levels between the X chromosome and the autosomes for each group were statistically different (*Supplementary files 2* and *3*). In addition, a separate analysis was performed to identify differentially expressed genes between transgenic and wild-type siblings of the same sex (*Supplementary files 4* and *5*). A raw read counting matrix of the genes based on aforementioned BAM-formatted alignment files of all groups was generated using GenomicAlignments R package (*Lawrence et al., 2013*). We then used DESeq2 R package to estimate size factors and dispersion values for the groups according to the raw read counting matrix, and fitted a final generalized linear model using the size factors and dispersion values (*Love et al., 2014*), which gave estimates of log fold change for each gene (*Supplementary file 4*). The design formula for differential expression analysis was set to '~Group', where 'Group' was a column in the sample sheet indicating four groups as described above. As a result, a p value for log fold change of each gene between groups was reported. In addition, a Benjamini-Hochberg (BH) adjusted p value for each gene was calculated to report a false positive rate (FDR). To determine whether the number of up-regulated genes was significantly higher for the X chromosome compared to the autosomes, a Pearson's Chi-square test was applied. A Chi-square test was also used to assess whether the chromosomal distribution of the 382 upregulated X chromosome genes showed apparent bias (*Figure 4*).

## Acknowledgements

This work was supported by NIH grants AI105575 and AI121284 to ZT and by the Virginia Experimental Station. We thank Robert Harrell of the Insect Transgenic Facility at the University of Maryland for embryonic injections and for generating some of the transgenic lines. The X-linked CFP line was provided by Anthony James of the University of California, Irvine. We thank Clément Vinauger from the Virginia Tech Department of Biochemistry for help with statistical analysis.

## Additional information

### Funding

| Funder | Grant reference number | Author |
| --- | --- | --- |
| National Institutes of Health | AI105575 | Zhijian Jake Tu |
| National Institutes of Health | AI121284 | Zhijian Jake Tu |

The funders had no role in study design, data collection and interpretation, or the decision to submit the work for publication.

### Author contributions

Yumin Qi, Formal analysis, Investigation, Methodology, Writing—original draft; Yang Wu, Data curation, Software, Formal analysis, Investigation, Methodology, Writing—original draft; Randy Saunders, Investigation, Methodology; Xiao-Guang Chen, Supervision, Funding acquisition; Chunhong Mao, Data curation, Formal analysis, Investigation, Methodology; James Kite Biedler, Formal analysis, Validation, Investigation, Writing—original draft, Writing—review and editing; Zhijian Jake Tu, Conceptualization, Formal analysis, Supervision, Funding acquisition, Methodology, Project administration, Writing—review and editing

## Author ORCIDs

Yang Wu (ID) https://orcid.org/0000-0002-1207-5566
Zhijian Jake Tu (ID) http://orcid.org/0000-0003-4227-3819

## Ethics

Animal experimentation: This study was performed in strict accordance with the recommendations in the Guide for the Care and Use of Laboratory Animals of the National Institutes of Health. All of the animals were handled according to approved institutional animal care and use committee (IACUC) protocols (IACUC #16-067) of the Virginia Tech.

## Decision letter and Author response

Decision letter https://doi.org/10.7554/eLife.43570.022
Author response https://doi.org/10.7554/eLife.43570.023

## Additional files

### Supplementary files

• Supplementary file 1. Spearman correlation analyses of the two RNA-Seq experiments.
DOI: https://doi.org/10.7554/eLife.43570.012

• Supplementary file 2. Analyses of median FPKMs of X-linked genes vs. autosomal genes. This file shows results from analyses of pooled biological replicates.
DOI: https://doi.org/10.7554/eLife.43570.013

• Supplementary file 3. Analyses of median FPKMs of X-linked genes vs. autosomal genes of Guy1 transgenic and sibling wild type males and females: Individual replicates.
DOI: https://doi.org/10.7554/eLife.43570.014

• Supplementary file 4. DESeq2 output, overlap of differentially expressed genes, scaffold positions of up-regulated and other X-linked genes (This file includes five sheets of an Excel workbook, uploaded separately)
DOI: https://doi.org/10.7554/eLife.43570.015

• Supplementary file 5. Significantly upregulated and downregulated genes on chromosome X and autosomes between Guy1 transgenic and wild-type sibling males and females.
DOI: https://doi.org/10.7554/eLife.43570.016

• Transparent reporting form
DOI: https://doi.org/10.7554/eLife.43570.017

### Data availability

Data submitted to SRA, PRJNA503140 : The transcriptome of Guy1-transgenic Anopheles stephensi L1 instar (TaxID: 30069).

The following dataset was generated:

| Author(s) | Year | Dataset title | Dataset URL | Database and Identifier |
|---|---|---|---|---|
| Qi Y, Wu Y, Saunders R, Chen X, Mao C, Biedler JK, Tu Z | 2018 | The transcriptome of Guy1-transgenic Anopheles stephensi L1 instar | https://www.ncbi.nlm.nih.gov/bioproject/PRJNA503140 | NCBI Bioproject, PRJNA503140 |

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
