## [Decision Letter]

Thank you for submitting your article "*Guy1*, a Y-linked embryonic signal, regulates dosage compensation in *Anopheles stephensi* by increasing X gene expression" for consideration by *eLife*. Your article has been reviewed by Detlef Weigel as the Senior Editor, a Reviewing Editor, and three reviewers. The following individuals involved in review of your submission have agreed to reveal their identity: Flaminia Catteruccia (Reviewer #1); Christine M. Disteche (Reviewer #3).

The reviewers have discussed the reviews with one another and the Reviewing Editor has drafted this decision to help you prepare a revised submission

Summary:

This study by Tu and colleagues builds upon their prior finding published in *eLife* that showed *Guy1* as a strong candidate for a male-determining factor. In this Research Advance, they now demonstrate that *Guy1* regulates dosage compensation in the mosquito and that up-regulation of X chromosome genes underlies the mechanism of *Guy1*-conferred female lethality. All reviewers agreed that this is an important study that addresses fundamental questions regarding sex determination and dosage compensation in mosquitos.

Essential revisions:

A key concern is the exclusive reliance on mean gene expression (MGE) to generate major conclusions. The reviewers agreed that it is advisable to provide additional, independent validation (this does not mean simple repetition of the same experiments). One suggestion was to perform dsRNA injections (or an inducible CRISPRi system, although quite challenging) could be performed to target *Guy1* in males. If *Guy1* is responsible for dosage compensation, X-linked genes should decrease in expression. However, the reviewers noted that RNAi might not be feasible based on the authors' previous paper.

Therefore, the reviewers suggest the following:

1) If the authors can provide independent validation to their ideas, the reviewers would like to see such data incorporated.

2) Alternatively, recognizing the difficulty of such experiments, if no independent validation is feasible, reasonable arguments for their conclusion (without additional experimental data) would be acceptable. In such a case, depending on the strength of such arguments, it may be advisable to adjust the strength of the conclusion (weaken the statements, provide caveats etc.).

---

## [Author Response]

Essential revisions:A key concern is the exclusive reliance on mean gene expression (MGE) to generate major conclusions. The reviewers agreed that it is advisable to provide additional, independent validation (this does not mean simple repetition of the same experiments). One suggestion was to perform dsRNA injections (or an inducible CRISPRi system, although quite challenging) could be performed to target Guy1 in males. If Guy1 is responsible for dosage compensation, X-linked genes should decrease in expression. However, the reviewers noted that RNAi might not be feasible based on the authors' previous paper.Therefore, the reviewers suggest the following:1) If the authors can provide independent validation to their ideas, the reviewers would like to see such data incorporated.2) Alternatively, recognizing the difficulty of such experiments, if no independent validation is feasible, reasonable arguments for their conclusion (without additional experimental data) would be acceptable. In such a case, depending on the strength of such arguments, it may be advisable to adjust the strength of the conclusion (weaken the statements, provide caveats etc.).

We agree that determining the effect of successful *Guy1* inhibition in males will provide additional evidence to the current study. However, as the reviewers pointed out, an inducible CRISPRi system is quite challenging and RNAi is equally difficult, given that Guy1 transcripts are only detected in a narrowly restricted time window during the early embryonic stage. Even if RNAi knockdown of *Guy1* was feasible, it would still be very difficult to identify the surviving individuals in which *Guy1* had been successfully knocked down, again because *Guy1* transcripts disappear regardless of treatments after 12-16 hours post egg deposition. Therefore, we concentrated on evaluating the effect of ectopic expression of *Guy1* on the transcription of X-linked genes in females. To rule out position effect, we used two transgenic lines with independent insertion sites, *nGuy1_1* and *nGuy1_2* (Criscione et al., 2016), both of which express *Guy1* from its native promoter. In both lines, we showed that the median level gene expression (MGE) of X-linked genes is significantly higher than autosomal genes in *Guy1*-transgenic females while there is no significant difference in MGE between X and autosomal genes in wild type females. Moreover, we did not exclusively rely on median gene expression (MGE). We also showed that *Guy1* significantly up-regulates 580 (in line *nGuy1_2*) and 475 (in line *nGuy1_1*) of the 996 X-linked genes in transgenic females, suggesting a chromosome-wide effect (Figure 2, Figure 2—figure supplement 1A). Therefore, we found two separate lines of evidence supporting Guy1’s influence on the X chromosome in transgenic females that was not observed in other genotypes: (1) an increase in X chromosome MGE; and (2) a large increase in the proportion of upregulated genes on the X chromosome relative to the autosomes. Furthermore, the 382 X-linked genes that are up-regulated in both transgenic lines showed a chromosome-wide distribution (newly added Figure 4). Taken together, we presented strong evidence from two independent *Guy1*-expression lines that support the chromosome-wide up-regulation of X chromosome genes by *Guy1*. We respectfully suggest that the evidence sufficiently supports our conclusion that *Guy1* is responsible either directly or indirectly for increasing X gene expression, which underlies dosage compensation in *Anopheles stephensi*. However, we completely agree with the reviewers that it is also important to demonstrate a decrease in the expression of X-linked genes in males as a result of *Guy1* inhibition. Therefore, we added the following in the second paragraph of the Discussion section as necessary caveats: “However, it remains to be seen whether *Guy1* suppression in males will result in a decreased expression of X-linked genes. Challenges remain to achieving and verifying *Guy1* suppression as *Guy1* transcription is narrowly restricted to the early embryonic stage”. We also modified the Abstract accordingly.